# Trends of Anogenital Warts: A 32-Year Retrospective Observational Study (Italy, 1991–2022)

**DOI:** 10.3390/jcm14113962

**Published:** 2025-06-04

**Authors:** Eugenia Giuliani, Maria Gabriella Donà, Mauro Zaccarelli, Christof Stingone, Laura Gianserra, Stella Capodieci, Valentina Cafaro, Chiara Fulgenzio, Alessandra Latini, Massimo Giuliani

**Affiliations:** 1STI/HIV Unit, San Gallicano Dermatological Institute IRCCS, 00144 Rome, Italy; eugenia.giuliani@ifo.it (E.G.); mauro.zaccarelli@ifo.it (M.Z.); christof.stingone@ifo.it (C.S.); laura.gianserra@ifo.it (L.G.); alessandra.latini@ifo.it (A.L.); massimo.giuliani@ifo.it (M.G.); 2Scientific Direction, San Gallicano Dermatological Institute IRCCS, 00144 Rome, Italy; stella.capodieci@ifo.it (S.C.); valentina.cafaro@ifo.it (V.C.); 3Hospital Pharmacy, San Gallicano Dermatological Institute IRCCS, 00144 Rome, Italy; chiara.fulgenzio@ifo.it

**Keywords:** anogenital warts, condylomata acuminata, human papillomavirus, sexually transmitted diseases, trend, HPV, vaccines

## Abstract

**Background**: Anogenital warts (AGWs) represent the most common clinical manifestation of human papillomavirus (HPV) infection. The analysis of AGW time trends helps to investigate possible changes over time and monitor the impact of HPV vaccines on HPV-associated morbidity. **Methods**: AGWs diagnosed from 1991 to 2022 at a centre belonging to the Italian STI Surveillance Network were analysed in terms of their numbers and the socio-behavioural characteristics of the patients and then compared over the course of three decades. The time trends, overall and by transmission category, were investigated by joinpoint analysis. **Results**: In total, 9781 AGWs were diagnosed [61.9% in men who have sex with women (MSW)]. Individuals aged 25–34 years (36.9%), those with one recent partner (52.0%), and those with no previous STIs (87.8%) accounted for the majority of diagnoses. The HIV prevalence among individuals tested for HIV was 136/3098 (4.4%, 95% CI: 3.7–5.2). AGW diagnoses showed a mild decrease from 1991 to 2004, followed by a significant increase up to 2013 (the highest peak over the entire study period) and a significant decline thereafter, until the most recent years. During 2011–2022, diagnoses in women who have sex with men (WSM) decreased for those aged up to 24 years—the age class that could potentially have included women vaccinated against HPV. **Conclusions**: AGWs have declined in recent years. The decrease in the proportion of diagnoses in young WSM may reflect the female immunisation campaign in Italy. Nonetheless, there is still a need to promote vaccine uptake to reduce the AGW burden.

## 1. Introduction

Anogenital warts (AGWs) are the most common sexually transmitted infection (STI) reported by STI surveillance systems in western countries. Every year, between 600,000 and 800,000 individuals in Europe are diagnosed with AGWs [1]. Reliable European data on the prevalence of AGWs are available only in England and Finland, where notifications are mandatory. Data from the Italian STI Sentinel Surveillance System confirm that AGWs represent the most widespread STI in Italy and showed an increase from 2004 to 2015 [2]. In Italy, an incidence of 1–5 per 1000 individuals has been estimated [3], and approximately 70,000 new cases per year are diagnosed among women aged between 15 and 64 years [4]. These data suggest that between 60,000 and 300,000 individuals seek specialised care for AGWs annually.

More than 200 human papillomavirus (HPV) genotypes have been officially established [5]. Around 30 genotypes are mucosotropic and are distinguished as high-risk and low-risk HPVs according to their carcinogenic potential [6]. Twelve HPVs (including HPVs 16 and 18) are classified into Group 1, since there is sufficient evidence of their carcinogenicity in humans. Indeed, they are involved in the development of pre-neoplastic and neoplastic lesions of the anogenital tract, as well as of the head and neck region [6]. HPVs 6 and 11 are instead classified into Group 3, which includes agents not classifiable as to their carcinogenicity to humans. For these two genotypes, there is little to no mechanistic evidence that they can contribute to carcinogenesis in humans. Instead, they are a common cause of AGWs, being responsible for more than 90% of cases [7,8]. These two HPV genotypes, in the form of virus-like particles, are included in quadrivalent and nonavalent HPV vaccines, and a great number of studies have shown a drastic decrease in AGWs in vaccinated adolescents [9,10]. Given the effectiveness of vaccination in decreasing the AGW incidence, it is important to clarify whether and how the epidemiology of this STI has changed over time in order to identify where interventions are still needed, such as campaigns to increase AGW awareness and maximise participation in HPV vaccination programs.

The analysis of standardised clinical and demographic data from a large case series collected over long periods provides pivotal information on the characteristics of individuals with a specific STI. In the present study, we evaluated the time trends of AGWs diagnosed over the last three decades at the largest STI centre in Rome, Italy. We also assessed the socio-behavioural variables of the patients and their possible changes over time.

## 2. Materials and Methods

### 2.1. Study Site and Population

All AGWs diagnosed among the attendees of the STI/human immunodeficiency virus (HIV) centre of the San Gallicano Dermatological Institute (IRCCS; Rome, Italy) between January 1991 and December 2022 were retrieved from the medical archives of the centre. This study site represents one of the public STI sentinel centres included in the clinical network of the STI Surveillance System in Italy and is located in a metropolitan area with the largest catchment area at the national level. During the study period, the characteristics of the centre did not change significantly, except during the COVID-19 pandemic. For the rest of the study period, no changes were introduced in terms of structure, process, and outcomes according to the Donabedian criteria [11] (i.e., number of consultation rooms, operators, opening days or hours, case definition, treatment options, or costs). Over the years, the diagnosis of AGWs has followed the Guidelines for the Management of AGWs of the International Union against Sexually Transmitted Infections (IUSTI)—Europe [12]. According to these guidelines, AGW diagnosis is clinically based. Biopsy is not necessary and is recommended only in cases of diagnostic uncertainty or suspicion of pre-cancer or cancer. In addition, HPV detection or typing is not recommended since it does not influence management. All physicians at the centre were expert dermatologists trained in STI management and control. All patients with AGWs underwent comprehensive anamnestic data collection and direct examination of the genital and perianal areas using a light lamp. Written informed consent for routine clinical examination and treatment was obtained from all patients.

### 2.2. Statistical Analyses

Analyses were performed for the overall study population and three study groups stratified according to the type of sexual partner. Men who reported having sex with women exclusively were classified as men who have sex with women (MSW). None of the women had ever had sex with other women, so they were classified as women who have sex with men (WSM), whereas men who reported having sex with men exclusively or with both men and women were classified as men who have sex with men (MSM). Summary statistics, i.e., proportions, medians, and interquartile ranges, were used to describe the study groups. Median values were compared using the Mann–Whitney and Kruskal–Wallis tests, as appropriate. The distributions of categorical variables in the study groups were compared using the chi-squared test. For the analysis, only individuals with a documented diagnosis of HIV were considered as living with HIV.

The joinpoint trend analysis software, version 5.4.0 (National Cancer Institute, Bethesda, MD, USA), was used to evaluate the temporal trends, setting a maximum number of two joinpoints. *p*-values < 0.05 were considered statistically significant. The remaining statistical analyses were conducted using the MedCalc^®^ Statistical Software, version 23.2.1 (MedCalc Software, Ltd., Ostend, Belgium; https://www.medcalc.org), and STATA version 15 (Stata Corporation, College Station, TX, USA).

## 3. Results

### 3.1. Study Population

During the entire study period, our STI/HIV centre diagnosed 9781 cases of AGWs. Diagnosis was clinically based for almost all cases. Histopathological evaluation was required to confirm the diagnosis for 176 cases (1.8%). The sociodemographic and behavioural characteristics of the study subjects are presented in Table 1.

The majority of AGWs were diagnosed in MSW (6054 cases; 61.9%), followed by WSM (2351 cases; 24.0%) and MSM (1376 cases; 14.1%). At diagnosis, MSW were significantly older than both WSM and MSM.

Stratifying the overall population by age class, AGWs were more frequent in individuals aged 25–34 years (36.9%). This was also confirmed in each group, particularly in MSM (43.5%). Almost all patients were Italian (91.9%), around half declared having one partner (52.0%), and the majority reported no previous STI diagnosis (87.8%). More than two-thirds of the patients had not been tested for HIV at the time of AGW diagnosis (68.3%). Considering the patients aware of their HIV serostatus, 136/3098 (4.4%; 95% CI: 3.7–5.2) were living with HIV. MSM showed a higher prevalence of HIV infection than MSW (74/691, 10.7% vs. 48/1745, 2.8%, *p* < 0.0001) Statistically significant differences were observed between WSM and MSW for most of the other descriptive characteristics, as well as between MSW and MSM.

### 3.2. Time Trend Analysis

The annual number of AGWs diagnosed in the overall study population from 1991 to 2022, together with the joinpoint analysis, is shown in Figure 1. The trend over these 32 years showed an initial decrease from 1991 to 2004, when the lowest number of annual diagnoses was recorded, followed by an initial peak in 2008, which showed a four-fold increase in annual cases compared with 2004. A decline in the number of diagnoses was observed from 2008 to 2010. From 2010, the trend saw a new rapid increase until 2013, which showed the highest peak during the entire study period. Subsequently, a progressive decrease was observed. Using joinpoint analysis to test whether the changes in trend were statistically significant, 2004 and 2013 were identified as the time points in which the trend significantly changed. The trend, in fact, showed a mild decrease from 1991 to 2004, followed by a significant increase until 2013 (slope: 48.30, 95% CI: from 19.37 to 77.23; *p* = 0.0049) and finally a sharp decline until the end of the observation period (slope: −48.22, 95% CI: from −58.70 to −37.74, *p* < 0.0001) (Table A1).

Trends in the annual AGWs in the three study groups are shown in Figure 2. The fitted curves highlighted increasing trends between 2004 and 2013 in all three groups, although statistical significance was reached only for MSW (*p* = 0.0031) and WSM (*p* = 0.0122) (Table A2). Subsequently, a significant decline was observed for all three risk groups, with the greatest change in MSW (slope: −31.04, 95% CI: from −37.57 to −24.50; *p* < 0.0001) (Table A2).

### 3.3. Sociodemographic and Behavioural Variables of Patients with AGWs by Period of Diagnosis

To assess whether the sociodemographic and behavioural characteristics of patients with AGWs changed over time, we analysed all study variables in the overall population by decade of diagnosis (Table 2).

The median age of the patients progressively increased, from 30 years in 1991–2000 to 37 years in 2011–2022 (*p* < 0.000001). After stratifying the patients by age class, it emerged that individuals aged 25–34 years accounted for the highest number of cases in all decades, but their relative contributions tended to decrease over time. In contrast, the number of AGWs diagnosed in patients older than 44 years more than doubled from 1991–2000 (13.2%) to 2011–2022 (30.4%). MSW accounted for the majority of the diagnoses in all three decades. Over time, the educational level of patients increased, the number of recent sexual partners decreased, and the proportion of those reporting consistent condom use increased. Previous STIs and injection drug use have been reported less frequently over time. Furthermore, the prevalence of HIV infection among patients with a known HIV status tended to decrease significantly over the study decades, from 6.7% (95% CI: 5.0–8.7) in 1991–2000 to 3.5% (95% CI: 2.6–4.5) in the last decade.

Table 3 shows that the patients’ median age increased over time in all study groups, although this increase was less pronounced for MSM.

Among WSM, the majority of the AGWs were diagnosed in those aged 25–34 in all three decades, but the relative contribution of this age class tended to decrease, as well as that of WSM <24 years old. Differently, among MSW, those aged over 44 years contributed to the highest number of diagnoses in the last decade, outnumbering the AGW cases observed in MSW aged 25–34 years.

## 4. Discussion

In our study, we examined the trends of AGWs diagnosed over a period of 32 years at our STI/HIV centre, which has the largest catchment area in Central and Southern Italy. We also evaluated possible changes over time in the sociodemographic and behavioural characteristics of the patients. Overall, we observed that the majority of diagnoses were among MSW, followed by WSM and MSM. Our findings suggest a 10:1 risk for AGW acquisition between heterosexual individuals and MSM. This ratio is far from that observed for other STIs and seems to reflect the proportion of MSM in the sexually active population [13]. It must be considered that the substantially higher contribution of men over WSM to our AGW cases is likely due to the fact that the attendees of our STI/HIV centre are mostly men, whereas women with HPV-related diseases are mostly referred to gynaecologists. We also observed that the majority of diagnoses were found in subjects aged 25–34 years, irrespective of sex and transmission category. Our data are consistent with those reported by the HERCOLES study in Portugal, which found the majority of AGWs among MSW and individuals aged 25–34 years [14], and those of a U.S. study that collected data from 40 STI clinics and found the large majority of diagnoses in MSW, followed by WSM and MSM [15]. Notably, the distribution of AGW diagnoses in our study population is similar to that reported by the Sentinel Surveillance System of the Italian National Health Institute during 1991–2021 (57.7%, 28.1%, and 14.2% in MSW, WSM, and MSM, respectively) [2]. However, while they reported that AGW diagnoses were more frequent in subjects aged 15–24 years, our STI/HIV centre recorded the highest number of diagnoses among older subjects.

The overall time trend of AGW diagnoses partially overlaps with that reported by the Italian Sentinel System [2]. A significant increase in diagnoses has been observed since 2004, with an initial, more modest peak in 2008 and a second peak in 2013, the highest in the entire study period. The reasons underlying these changes in the trend could be numerous, heterogeneous, and complex to define. The increase in the number of diagnoses could be due to changes in health-seeking behaviours, a greater STI risk (e.g., associated with the increasing use of dating apps in the last few years), or increased awareness of HPV-related diseases due to HPV vaccination campaigns, among others.

The 2013 peak was followed by a significant decline until the end of the observation period. Notably, the last few years of our study period saw the COVID-19 pandemic. The impact of the pandemic on STI/HIV services has been considerable and has affected STI diagnosis [16]. The substantial decrease observed in 2020 from the previous year was due to changes in access to our service. In fact, our centre remained open even during lockdowns, but consultations were limited to patients with bacterial STI-related complaints who needed treatment and key at-risk populations, such as MSM [17,18].

It is worth noting that the overall AGW trend was mainly sustained by MSW, in whom the diagnoses showed the same trend as the overall population, with a significant increase between 2004 and 2013 and a significant decline thereafter. The trend in MSM also showed an increase between 2004 and 2013, but this was not significant. Among MSM, AGWs showed the highest peak in earlier years compared with MSW and WSM, consistent with the Sentinel System data [2]. The decline since 2013 was significant also in MSM. Notably, our centre not only routinely recommends HPV vaccination to MSM, but also has an agreement with the local health authority (ASL) to allow vaccination without an appointment for MSM users from the centre.

Based on our observations, AGWs appear to differ from other STIs (i.e., syphilis, gonorrhoea, and HIV) in terms of their burden, risk factors, and core population. Early syphilis and gonorrhoea are the effective biological outcomes of recent risky sexual behaviour and thus can be used to monitor changes in the behaviour of sexually active populations. They are also responsible for sudden outbreaks in specific at-risk groups, such as MSM, commercial sex workers, and small marginal communities [2,19]. In contrast, our findings show that AGWs seem to occur particularly in heterosexual individuals with a low number of recent sexual partners (over half of the patients reported having only one partner) and no history of other STIs. In addition, approximately one-third of the patients reported that they always used condoms. Condoms are not fully protective against genital HPV infections, and their effectiveness against HPV appears to be lower than for other STIs [20,21,22]. All these observations seem to outline a different epidemiology for AGWs compared to that of other viral and bacterial STIs.

Generally, people living with HIV are at an increased risk of HPV acquisition, prevalence, and persistence [23,24,25]. In this population, AGWs differ from those diagnosed in HIV-negative individuals in terms of their natural history (e.g., tendency to recur), occurrence at unusual anatomical sites, and difficulty in treatment [26]. In our study, the HIV prevalence among all patients with AGWs tested for HIV was slightly over 4%, and it was the highest among MSM (over 10%). Our data are in line with those reported by the Italian Sentinel System [2]. Notably, the HIV prevalence among patients with AGWs between 1991 and 2021 was close to that reported for other patients with STIs but was substantially lower than that observed in individuals with infectious syphilis [2]. The HIV prevalence among our patients with AGWs declined significantly over time, which aligns with the national sentinel data for these patients and data on the Italian general population [2].

Two prophylactic vaccines are available for the prevention of HPV 6 and HPV 11 infections, which have close to 100% efficacy against genital warts. In Italy, a free and active national vaccination program for 12-year-old girls began in 2007/2008 [27]. This may have contributed to the trend in AGWs recorded among WSM, for whom a progressive decline in diagnoses has been observed since 2014. Given that the HPV vaccination campaign targeted 12-year-old girls starting from the 1997 birth cohort, vaccine-eligible girls could not be represented among our patients with AGWs in the first two decades. If we consider their sexual debut to be approximately five to six years from vaccination, the first cohorts of vaccinated girls likely began engaging in sexual activity at the beginning of the third study decade, when we began to observe a decrease in AGW diagnoses. In this decade, the diagnoses among WSM decreased for the age class of up to 24 years, which could potentially have included vaccinated women. This finding supports the hypothesis of the first visible effects of the female immunisation campaign on the AGW trend, although with suboptimal regional coverage. In the Lazio region, where our STI/HIV centre is located, the HPV vaccination coverage (last dose) for females from the 1997 birth cohort and onwards was between 66% and 76% in the last decade of observation of the present study [28]. The active and free HPV vaccine offer was extended to 12-year-old boys only in 2017 according to the National Immunization Plan 2017–2019 [29]. Therefore, we were unlikely to see the direct impact of male vaccination on our AGW diagnoses, since the start of the national campaign targeting boys was close to the end of the study period. Nonetheless, MSW aged up to 24 years experienced the largest decline in AGW diagnoses in the last decade compared with previous years, suggesting a possible herd effect derived from immunisation in girls. It is worth noting that several studies have investigated the therapeutic potential of prophylactic HPV vaccines, and some investigators have shown that the vaccination of AGW patients is associated with a reduction in subsequent lesions, as reviewed by Reuschenbach and collaborators [30]. However, evidence in this regard is not conclusive, since the sample size is often limited and statistical significance is not reached in the majority of studies. A recent cancer-based study on over 2 million individuals showed that the quadrivalent HPV vaccine is not protective against a second AGW episode when administered after the first episode [31]. A meta-analysis also evidenced that the rate of AGW recurrence found in two randomised controlled trials (RCT) was similar between the vaccine group and the control group [32]. Other RCTs are ongoing and might shed more light on the impact of the HPV vaccine on AGW recurrence.

As previously mentioned, AGW diagnosis was clinically based, and neither histopathological examination nor HPV testing was performed in our case series. Although over 90% of AGWs are caused by HPVs 6 and 11, other HPVs, including high-risk types, can be detected in such lesions [7,8]. A substantial reduction in prevalent HPV 6 and 11 infections (up to 90%) has been observed in the vaccination era [10], and changes in their prevalence in AGWs have been described following the implementation of the prophylactic vaccination [33].

Our study has a few strengths. First, it is the only study conducted in Italy that describes the trends in AGW diagnoses in an STI/HIV centre over a period of more than 30 years, during which HPV vaccination programs were implemented in both girls and boys. Second, the study provides indirect evidence of the effectiveness of vaccination in preventing AGWs. Finally, the characteristics of our centre’s operations, the diagnostic criteria, and the sociodemographic and behavioural information collected from each patient remained stable throughout the study period, thus allowing us to describe changes in the characteristics of patients with AGWs over time. The only changes were introduced during the COVID-19 pandemic, as discussed. Since we cannot exclude the fact that multiple diagnoses in some patients may have contributed to our case series over different years, this is a limitation of our study. A further limitation is represented by the lack of data on vaccination coverage in our study population, either at the eligible age for free and active vaccination or in adulthood. Finally, we had no data on the prevalence of the different HPV types in our patients, since AGWs were not tested for HPV.

## 5. Conclusions

Nearly 10,000 AGW cases have been diagnosed in our STI/HIV centre over the last 32 years, most of which are in MSW. Although most patients with AGWs were aged 25–34 years, the median age tended to increase significantly over time. Overall, the highest peak in diagnoses was observed in 2013, with a subsequent decline. In the coming years, the monitoring of AGW diagnoses could help to estimate the impact of HPV vaccination. Since vaccination coverage remains suboptimal, especially among males (around 50% starting from the 2006 birth cohort, the first to be offered free vaccination) [34], there is still a need to promote vaccine uptake to reduce AGWs, which is an achievable result, as real-world data have clearly demonstrated [10,35].

## Figures and Tables

**Figure 1 jcm-14-03962-f001:**
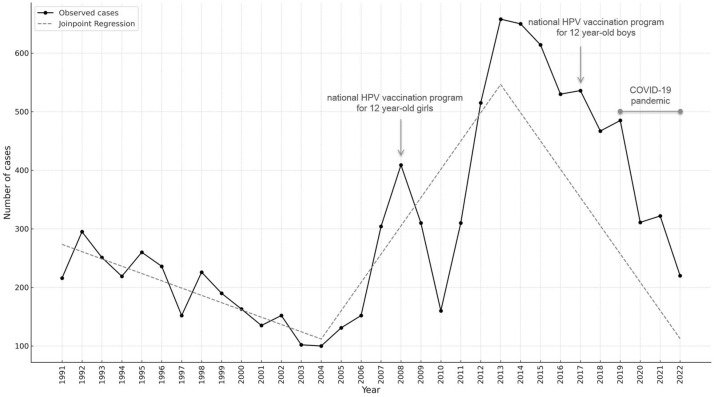
Number of anogenital warts by calendar year among the attendees of the STI/HIV centre of the San Gallicano Dermatological Institute (IRCCS; Rome, Italy) from 1991 to 2022; the fitted curve resulting from the joinpoint analysis is shown; years of implementation of free HPV vaccination for adolescent girls and boys are indicated, along with the COVID-19 pandemic years.

**Figure 2 jcm-14-03962-f002:**
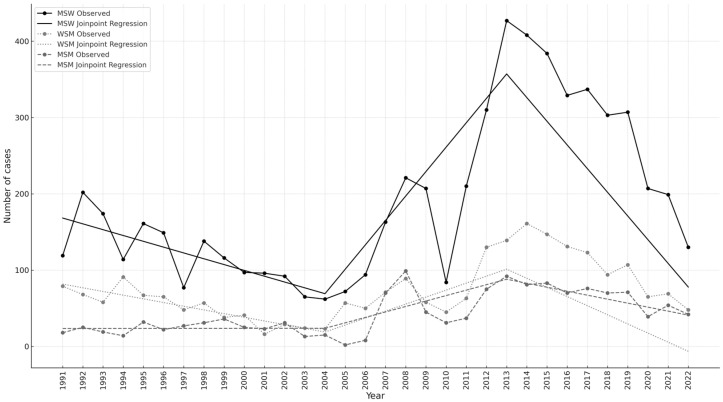
Number of anogenital warts by calendar year in women who have sex with men (WSM), men who have sex with women (MSW), and men who have sex with men (MSM) from 1991 to 2022; the fitted curves resulting from the joinpoint analysis are also shown.

**Table 1 jcm-14-03962-t001:** Sociodemographic, behavioural, and clinical variables of 9781 patients with anogenital warts by transmission category, 1991–2022.

Variable	Total n = 9781	WSM n = 2351	MSW n = 6054	MSM n = 1376	*p*-Value ^1^	*p*-Value ^2^
**Age, median (IQR)**	34 (27–44)	32 (26–42)	35 (28–45)	32 (27–41)	**<0.0001**	**<0.0001**
	**n (%)**					
**Age class, years**					**<0.0001**	**<0.0001**
Up to 24	1349 (13.8)	453 (19.3)	691 (11.4)	205 (14.9)		
25–34	3605 (36.9)	864 (36.8)	2143 (35.4)	598 (43.5)		
35–44	2521 (25.8)	553 (23.5)	1641 (27.1)	327 (23.8)		
Over 44	2306 (23.6)	481 (20.5)	1579 (26.1)	246 (17.9)		
**Education**					**<0.0001**	**<0.0001**
Ungraduated	7971 (81.5)	1883 (80.1)	5064 (83.6)	1024 (74.4)		
Graduated	1547 (15.8)	372 (15.8)	833 (13.8)	342 (24.9)		
Unknown	263 (2.7)	96 (4.1)	157 (2.6)	10 (0.7)		
**Nationality**					**<0.0001**	**0.016**
Italian	8987 (91.9)	2083 (88.6)	5646 (93.3)	1258 (91.4)		
Not Italian	794 (8.1)	268 (11.4)	408 (6.7)	118 (8.6)		
**No. recent partners ^3^**					**<0.0001**	**<0.0001**
0	888 (9.1)	316 (13.4)	506 (8.4)	66 (4.8)		
1	5085 (52.0)	1467 (62.4)	3317 (54.8)	301 (21.9)		
2	1893 (19.4)	422 (17.9)	1237 (20.4)	234 (17.0)		
3	786 (8.0)	86 (3.7)	475 (7.8)	225 (16.4)		
>3	1129 (11.5)	60 (2.6)	519 (8.6)	550 (40.0)		
**Condom use**					**<0.0001**	**<0.0001**
Never	3949 (40.4)	1239 (52.7)	2472 (40.8)	238 (17.3)		
Not always	2633 (26.9)	475 (20.2)	1593 (26.3)	565 (41.1)		
Always	3159 (32.3)	636 (27.1)	1978 (32.7)	545 (39.6)		
Unknown	40 (0.4)	1 (0.04)	11 (0.2)	28 (2.0)		
**Injection drug use**					0.26	**0.01**
No	9562 (97.8)	2305 (98.0)	5899 (97.4)	1358 (98.7)		
Yes	180 (1.8)	38 (1.6)	129 (2.1)	13 (0.9)		
Unknown	39 (0.4)	8 (0.3)	26 (0.4)	5 (0.4)		
**History of STI**					**0.0003**	**<0.0001**
No	8588 (87.8)	2175 (92.5)	5428 (89.7)	985 (71.6)		
Yes	1171 (12.0)	172 (7.3)	610 (10.1)	389 (28.3)		
Unknown	22 (0.2)	4 (0.2)	16 (0.3)	2 (0.1)		
**HIV status**					0.57	**<0.0001**
Negative	2962 (30.3)	648 (27.6)	1697 (28.0)	617 (44.8)		
Positive	136 (1.4)	14 (0.6)	48 (0.8)	74 (5.4)		
Unknown	6683 (68.3)	1689 (71.8)	4309 (71.2)	685 (49.8)		

^1^ WSM vs. MSW; ^2^ MSW vs. MSM; ^3^ during the previous 6 months. WSM, women who have sex with men; MSW, men who have sex with women; MSM, men who have sex with men; IQR, interquartile range; STI, sexually transmitted infection. Statistically significant differences are indicated in bold.

**Table 2 jcm-14-03962-t002:** Sociodemographic, behavioural, and clinical variables of 9781 patients with anogenital warts by decade of diagnosis.

Variable	1991–2000 n = 2208	2001–2010 n = 1955	2011–2022 n = 5618	*p*-Value
**Age, median (IQR)**	30 (25–37)	33 (26–41)	37 (29–47)	**<0.000001**
	**n (%)**			
**Age class, years**				**<0.0001**
Up to 24	460 (20.8)	348 (17.8)	541 (9.6)	
25–34	1021 (46.3)	736 (37.6)	1848 (32.9)	
35–44	436 (19.7)	565 (28.9)	1520 (27.1)	
Over 44	291 (13.2)	306 (15.7)	1709 (30.4)	
**Transmission category**				**<0.0001**
WSM	612 (27.7)	462 (23.6)	1277 (22.7)	
MSW	1347 (61.0)	1156 (59.1)	3551 (63.2)	
MSM	249 (11.3)	337 (17.2)	790 (14.1)	
**Education**				**<0.0001**
Ungraduated	2019 (91.5)	1438 (73.6)	4514 (80.3)	
Graduated	182 (8.2)	266 (13.6)	1099 (19.6)	
Unknown	7 (0.3)	251 (12.8)	5 (0.09)	
**Nationality**				**<0.0001**
Italian	2055 (93.1)	1743 (89.2)	5189 (92.4)	
Not Italian	153 (6.9)	212 (10.8)	429 (7.6)	
**No. recent partners ^1^**				**<0.0001**
0	135 (6.1)	138 (7.1)	615 (10.9)	
1	1133 (51.3)	830 (42.4)	3122 (55.6)	
2	429 (19.4)	516 (26.4)	948 (16.9)	
3	213 (9.7)	241 (12.3)	332 (5.9)	
>3	298 (13.5)	230 (11.8)	601 (10.7)	
**Condom use**				**<0.0001**
Never	1622 (73.5)	788 (40.3)	1539 (27.4)	
Not always	451 (20.4)	804 (41.1)	1378 (24.5)	
Always	135 (6.1)	363 (18.6)	2661 (47.4)	
Unknown	0 (0.0)	0 (0.0)	40 (0.7)	
**Injection drug use**				**<0.0001**
No	2088 (94.6)	1910 (97.7)	5564 (99.0)	
Yes	106 (4.8)	43 (2.2)	31 (0.6)	
Unknown	14 (0.6)	2 (0.1)	23 (0.4)	
**History of STI**				**<0.0001**
No	1676 (75.9)	1646 (84.2)	5266 (93.7)	
Yes	526 (23.8)	308 (15.8)	337 (6.0)	
Unknown	6 (0.3)	1 (0.05)	15 (0.3)	
**HIV status**				**<0.0001**
Negative	783 (35.5)	659 (33.7)	1520 (27.0)	
Positive	56 (2.5)	25 (1.3)	55 (1.0)	
Unknown	1369 (62.0)	1271 (65.0)	4043 (72.0)	

^1^ During the previous 6 months. WSM, women who have sex with men; MSW, men who have sex with women; MSM, men who have sex with men; IQR, interquartile range; STI, sexually transmitted infection. Statistically significant differences are indicated in bold.

**Table 3 jcm-14-03962-t003:** Median age of patients with anogenital warts for each study group and distribution of WSM, MSW, and MSM patients by age class in the three decades.

	Decade of Diagnosis			*p*-Value
	1991–2000	2001–2010	2011–2022	
WSM	n = 612	n = 462	n = 1277	
**Median age (IQR)**	28 (23–35)	30 (24–38)	37 (28–46)	**<0.000001**
**Age class, years**	**n (%)**			**<0.0001**
Up to 24	185 (30.2)	118 (25.5)	150 (11.7)	
25–34	265 (43.3)	181 (39.2)	418 (32.7)	
35–44	99 (16.2)	102 (22.1)	352 (27.6)	
Over 44	63 (10.3)	61 (13.2)	357 (28.0)	
**MSW**	**n = 1347**	**n = 1156**	**n = 3551**	
**Median age (IQR)**	31 (26–39)	35 (27–42)	38 (30–48)	**<0.000001**
**Age class, years**	**n (%)**			**<0.0001**
up to 24	239 (17.7)	159 (13.8)	293 (8.3)	
25–34	615 (45.7)	418 (36.1)	1110 (31.2)	
35–44	289 (21.5)	366 (31.7)	986 (27.8)	
over 44	204 (15.1)	213 (18.4)	1162 (32.7)	
**MSM**	**n = 249**	**n = 337**	**n = 790**	
**Median age (IQR)**	30 (26–35)	31 (26–38)	33 (28–44)	**<0.000001**
**Age class, years**	**n (%)**			**<0.0001**
up to 24	36 (14.5)	71 (21.1)	98 (12.4)	
25–34	141 (56.6)	137 (40.6)	320 (40.5)	
35–44	48 (19.3)	97 (28.8)	182 (23.0)	
over 44	24 (9.6)	32 (9.5)	190 (24.1)	

WSM, women who have sex with men; MSW, men who have sex with women; MSM, men who have sex with men; IQR, interquartile range.

## Data Availability

The datasets analysed during the current study are available from the corresponding author on reasonable request.

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
