# Peer review of "Trends of Anogenital Warts: A 32-Year Retrospective Observational Study (Italy, 1991–2022)"

_jcm, 2025, doi:10.3390/jcm14113962_

Round 1

Reviewer 1 Report

Comments and Suggestions for Authors

The article “Trends of anogenital warts: a 30-year retrospective observational study (19991-2022)” describes the evolution of genital warts diagnosed over 30 years in an STD sentinel center in Rome, Italy.

It is a long period of observation, with a high number of cases and therefore the description of the characteristics of the total population is of scientific interest, as shown in table 1.

The time-trend analysis over decades is important because it can allow the indirect evaluation of the role of HPV vaccination in this population: the decrease in warts in patients over 24 and 25-34 years described in Table 1, can be explained by vaccination.

However, the fact that the temporal analysis by decades does not discriminate by age the transmission category reduced the understanding of the data.

It would be interesting to know how the number of WSM, MSW and MSM patients, respectively under 24 years old, 25-34, 35-44 and > 45 years old, behaves over the 3 decades. Since the HPV vaccine was introduced for girls at age 12 in 2008, the impact of this measure will be mainly on WSM up to age 24 and between 25-44 years.

It is also important to describe the HPV vaccination rate in the population in order to better contextualize the results.

Author Response

The article “Trends of anogenital warts: a 30-year retrospective observational study (19991-2022)” describes the evolution of genital warts diagnosed over 30 years in an STD sentinel center in Rome, Italy. It is a long period of observation, with a high number of cases and therefore the description of the characteristics of the total population is of scientific interest, as shown in table 1. The time-trend analysis over decades is important because it can allow the indirect evaluation of the role of HPV vaccination in this population: the decrease in warts in patients over 24 and 25-34 years described in Table 1, can be explained by vaccination.

Comment 1: However, the fact that the temporal analysis by decades does not discriminate by age the transmission category reduced the understanding of the data. It would be interesting to know how the number of WSM, MSW and MSM patients, respectively under 24 years old, 25-34, 35-44 and > 45 years old, behaves over the 3 decades. Since the HPV vaccine was introduced for girls at age 12 in 2008, the impact of this measure will be mainly on WSM up to age 24 and between 25-44 years.

Response 1: Data regarding the individual transmission categories stratified by age class are shown in Table 3, which, for some reason, was not displayed in the submitted version of the manuscript. We apologize for this inconvenience. This Table (page 8) shows that the majority of the AGWs in WSM were diagnosed in those aged 25-34 in all the three decades, although the relative contribution of this age class tended to decrease, as well as that of WSM <24 year-old. These results have been now described in the last paragraph of the Results (page 8, lines 201-205).

Comment 2: It is also important to describe the HPV vaccination rate in the population in order to better contextualize the results.

Response 2: Unfortunately, we do not have data about the HPV vaccination rate in our population, and we have now included this aspect among the study limits (page 11, lines 326-329). Nonetheless, we have now added data on the vaccination program coverage for the relevant birth cohorts in the Lazio region, where our STI/HIV Centre is located (page 10, lines 289-291).

Reviewer 2 Report

Comments and Suggestions for Authors

Thank you for the opportunity to review this work. I want to congratulate the authors on several aspects. Firstly, the form of presentation of the law deserves praise - it is legible, systematic, clear and correctly written. Secondly, it addresses an aspect of AGW that is not so widely discussed in the literature, and the authors obtained data from 30 years, which is a long area of observation. Thirdly, the authors took into account the impact of HPV vaccinations on the frequency of warts in the population. However, I would like to raise one issue about vaccination. The authors talk about preventive population programs for children and 12-year-old girls. However, do you know what part of the population was vaccinated against HPV in adulthood? It is already known that the impact of vaccination in HPV (+) individuals can modify the course of the disease, both HSIL of the cervix and the frequency of condyloma recurrences. Please raise this aspect. My second comment is to please attach information on the frequency of HPV genotypes in Italy.
Were the warts examined histopathologically, or were they only diagnosed clinically? LSIL or HSIL can develop within the warts, similarly to the cervix. Please refer to the following paper on genotyping in LSIL lesions and try to answer the question of what HPV genotypes were most common in your patients. 
Kedzia W, Józefiak A, Pruski D, Rokita W, Marek S. Genotypowanie wirusów brodawczaka ludzkiego u kobiet z CIN 1 [Human papilloma virus genotyping in women with CIN 1]. Ginekol Pol. 2010 Sep;81(9):664-7. Polish. PMID: 20973202.
Please make these two minor changes

Author Response

Comment 1: Thank you for the opportunity to review this work. I want to congratulate the authors on several aspects. Firstly, the form of presentation of the law deserves praise - it is legible, systematic, clear and correctly written. Secondly, it addresses an aspect of AGW that is not so widely discussed in the literature, and the authors obtained data from 30 years, which is a long area of observation. Thirdly, the authors took into account the impact of HPV vaccinations on the frequency of warts in the population. However, I would like to raise one issue about vaccination. The authors talk about preventive population programs for children and 12-year-old girls. However, do you know what part of the population was vaccinated against HPV in adulthood? It is already known that the impact of vaccination in HPV (+) individuals can modify the course of the disease, both HSIL of the cervix and the frequency of condyloma recurrences. Please raise this aspect.

Response 1: We thank the reviewer for appreciating our study. Regarding the vaccination in adulthood, unfortunately, we do not have these data, so we have acknowledged this limit in the respective section (page 11, lines 326-329). Regarding condyloma recurrence, we have now incorporated data from a review, a register-based study and a meta-analysis that evaluated the available literature on this topic (page 10, lines 298-308).

Comment 2: My second comment is to please attach information on the frequency of HPV genotypes in Italy. Were the warts examined histopathologically, or were they only diagnosed clinically? LSIL or HSIL can develop within the warts, similarly to the cervix. Please refer to the following paper on genotyping in LSIL lesions and try to answer the question of what HPV genotypes were most common in your patients. Kedzia W, Józefiak A, Pruski D, Rokita W, Marek S. Genotypowanie wirusów brodawczaka ludzkiego u kobiet z CIN 1 [Human papilloma virus genotyping in women with CIN 1]. Ginekol Pol. 2010 Sep;81(9):664-7. Polish. PMID: 20973202.
Please make these two minor changes

Response 2: As per IUSTI guidelines, AGW diagnosis in clinically based and biopsy is not necessary for typical anogenital warts, although it is recommended if there is diagnostic uncertainty or suspicion of precancer or cancer. In addition, HPV detection or typing does not influence management and is not recommended. Since we follow IUSTI guidelines, as already mentioned, we have now included these details in the M&M section (page 3, lines 85-88). We have also added to the Results that, in a minor proportion of cases, histopathological evaluation was required to confirm the diagnosis (page 3, lines 115-116). We have also added that lack of data on the HPV genotypes present in our AGW cases may represent a further limitation of the study (page 11, lines 328-329). We have not included the suggested reference since our study concerns external ano-genital warts, not CIN lesions.

Reviewer 3 Report

Comments and Suggestions for Authors

          The authors analyzed the trends of anogenital wards (AGWs) in Italy during 30 years. They observed a decline in women presenting with this manifestation in the more recent period, probably associated with the beneficial effect of vaccination. Some concerns should be addressed before acceptance of this manuscript

  1. The authors mention two strengths and one limitation of theirs study. Another strength is providing indirect evidence, although limited since recent, of the effectiveness of vaccination for preventing AGWs. Another limitation, not mentioned, is the lack of information on the genotypes associated with these AGWs. Can the authors have access to at least of partial information on the main genotypes associated to these AGWs? Although the main genotypes would be 6 and 11, a shift in these genotypes may occur in the last years, because of vaccination, and this would be important to be monitored (see for example Magdaleno-Tapial J, et al. Changes in the Prevalence of Human Papillomavirus Genotypes in Genital Warts Since the Introduction of Prophylactic Vaccines. Actas Dermosifiliogr. 2022 Oct;113(9):T874-T880. doi: 10.1016/j.ad.2022.08.014.).
  2. A brief description of papillomavirus, its pathogenic associations (not only AGWs: high risk and low risk HPV) should be included.
  3. What is the reason of the increase in AGWs, particularly in women, in 2006, then the abrupt decrease in 2010, with an intense increase until 2013, and then a decline? The authors should describe better their data and propose probable reasons for the epidemiological behavior.
  4. Lines 244-245: a reference should be included.

Author Response

The authors analyzed the trends of anogenital wards (AGWs) in Italy during 30 years. They observed a decline in women presenting with this manifestation in the more recent period, probably associated with the beneficial effect of vaccination. Some concerns should be addressed before acceptance of this manuscript

Comment 1: The authors mention two strengths and one limitation of theirs study. Another strength is providing indirect evidence, although limited since recent, of the effectiveness of vaccination for preventing AGWs.

Response 1: Thank you for this suggestion. We have now included this further strength of the study (page 11, lines 319-320).

Comment 2: Another limitation, not mentioned, is the lack of information on the genotypes associated with these AGWs. Can the authors have access to at least of partial information on the main genotypes associated to these AGWs? Although the main genotypes would be 6 and 11, a shift in these genotypes may occur in the last years, because of vaccination, and this would be important to be monitored (see for example Magdaleno-Tapial J, et al. Changes in the Prevalence of Human Papillomavirus Genotypes in Genital Warts Since the Introduction of Prophylactic Vaccines. Actas Dermosifiliogr. 2022 Oct;113(9):T874-T880. doi: 10.1016/j.ad.2022.08.014.).

Response 2: We have now mentioned this further limitation of the study (page 11, lines 328-329). We have also added to the Discussion the possible change in the HPV types responsible for AGWs and included the suggested reference (page 11, lines 312-315).

Comment 3: A brief description of papillomavirus, its pathogenic associations (not only AGWs: high risk and low risk HPV) should be included.

Response 3: This has been included in the Introduction as suggested (page 2, lines 47-57).

Comment 4: What is the reason of the increase in AGWs, particularly in women, in 2006, then the abrupt

decrease in 2010, with an intense increase until 2013, and then a decline? The authors should describe better their data and propose probable reasons for the epidemiological behavior.

Response 4: We have speculated on the possible reasons underlying these changes (page 9, lines 231-236).

Comment 5: Lines 244-245: a reference should be included.

Response 5: We have now include the proper reference (page 10, line 278, ref. 27).

Round 2

Reviewer 3 Report

Comments and Suggestions for Authors

      The authors addressed satisfactorey the concerns.